# Types and anatomical locations of injuries among mountain bikers and hikers: A systematic review

Paul John Braybrook[1,2]*, Hideo Tohira[1,3], Tanya Birnie[1], Deon Brink[1,2], Judith Finn[1,2,3], Peter Buzzacott[1]

1 Prehospital Resuscitation and Emergency Care Research Unit, Curtin School of Nursing, Curtin University, Bentley, Western Australia, 2 St John Western Australia, Belmont, Western Australia, 3 Discipline of Emergency Medicine, Medical School, University of Western Australia, Crawley, Western Australia

* paul.braybrook@postgrad.curtin.edu.au

**Data Availability Statement:** All relevant data are within the manuscript and its Supporting Information files

## Abstract

### Background

Mountain biking and hiking continue to grow in popularity. With new participants to these sports, it is likely the number of injuries will increase. To assist medical personnel in the management of these patients we attempted to quantify the types and locations of injuries sustained by mountain bikers and hikers. Objective The objective of this systematic review is to identify the type and anatomical location of injuries for both mountain bikers and hikers.

### Methods

A systematic search was undertaken using CINAHL, Cochrane, ProQuest, PubMed and Scopus databases. Reviewers assessed the eligibility of articles by a title/abstract review and final full-text review. Studies were included if the types of injuries were reported by medical personnel and contained anatomical locations. Studies were excluded if it did not take place on a trail or if the injuries were self-reported. Risk of bias was assessed utilising the Joanna Briggs Institute (JBI) checklists for study quality. No meta-analysis or comparison between mountain bikers and hikers was possible due to the high heterogeneity of the definition of injury.

### Results

A total of 24 studies met the inclusion criteria, 17 covering mountain biking and 7 hiking. This represented 220,935 injured mountain bikers and 17,757 injured hikers. The most common type of injuries sustained by mountain bikers included contusions, abrasions and minor lacerations, which made up between 45–74% of reported injuries in studies on competitive racing and 8–67% in non-competitive studies. Fractures represented between 1.5–43% of all reported injuries. The most injured region was the upper limbs reported in 10 of 17 studies. For hikers the most common injuries included blisters and ankle sprains with blisters representing 8–33% of all reported injuries. The most common body location to be injured by hikers was a lower limb in all 7 studies.

**Funding:** The author(s) received no specific funding for this work.

**Competing interests:** The authors have declared that no competing interests exist.

**Abbreviations:** JBI, Joanna Briggs Institute; WHO, World Health Organisation; NCDs, Non communicable diseases; BMI, Body mass index; PRISMA, Preferred reporting items for systematic reviews and meta-analysis; ED, Emergency deportment; SD, Standard deviation.

## Conclusions

This is the first systematic review to report on the injury epidemiology of the two most common trail users; mountain bikers and hikers. For participants in both activities the majority of injuries were of minor severity. Despite this, the high proportions of upper limb fractures in mountain bikers and ankle sprains in hikers cannot be ignored.

## Trial registration

*Registration*: This systematic review was prospectively registered with the University of York PROSPERO database on the 12/4/2021 (CRD42021229623) https://www.crd.york.ac.uk/prospero/display_record.php?ID=CRD42021229623.

## Introduction

The World Health Organisation (WHO) state there are four main risk factors linked with non-communicable diseases (NCDs) worldwide, one being physical inactivity [1]. Coronary heart disease [2], colon cancer [3], breast cancer [4] and type two diabetes mellitus [5] have all been related to levels of inactivity and additionally, inactivity is strongly linked to obesity which is also considered a modifiable risk factor for a range of NCDs [6]. The WHO recommend children between 5–17 years old complete 60 minutes of moderate or vigorous activity each day and adults should undertake between 150 and 300 minutes of moderate physical activity per week [7].

Access to natural amenities such as trails is associated with increased physical activity levels and a reduction in obesity [8]. Both mountain biking and hiking utilise terrestrial trails and are self-propelled in that they do not utilise a motor or external means of propulsion such as a horse. The benefits of cycling are well reported and include improvements in cardiovascular health, body composition, blood lipids and an overall mortality decrease compared with non-cyclists [9–12]. Both hiking and mountain biking have also been shown to bring economic benefits to areas where the trails are present, due to increased tourism [13,14]. Additionally, economic benefits have been shown through the avoidance of sedentary living [15].

Acute traumatic injuries in mountain biking are reported to be predominantly lacerations and abrasions, with more serious injuries being mainly upper extremity fractures [16–18]. As mountain biking has evolved, bike design has changed and personal protective equipment use, including helmets, has increased. This has led to a decrease in previously seen head injuries and liver haematomas from poorly designed handlebar ends [19,20].

Hiking has also been shown to have benefits on cardiovascular fitness, especially for people who have metabolic syndrome where a course of regular hiking can reduce body mass index (BMI), blood pressure, mean arterial pressure, cholesterol and insulin resistance [21,22]. In addition to the many physical benefits offered by regular hiking, research into the mental health effects of hiking have shown benefits such as decreases in depression and reduced instances of suicidal ideation [23]. Studies of search and rescue events within US national parks found most incidents involved hikers [24–26]. Previous reports on the epidemiology of hiking injuries have investigated deaths while hiking in alpine environments, but little work exists examining injuries people sustain while hiking in non-mountainous regions [27].

Participation in hiking has increased in Australia over the past twenty years and it is estimated that around 7% of the current population participate in hiking each year [28].

Participation may be even higher as people who identify as completing recreational walking, 43% of the Australian population, may indeed be hiking. Whilst worldwide data on participation numbers in hiking are not available, data recorded by the US National Parks Service shows a generally increasing trend in visitor numbers, which may correspond to an overall increase in hikers in these areas [29].

This research will inform health providers and emergency medical services of likely injury types and anatomical locations from these activities. Further, it will assist government departments in planning trails in ways to minimise injuries.

## Aim

The aim of this research is to identify the type and bodily location of injuries in trail users. Specifically, the objective is to do this for both mountain bikers and hikers, as they are considered the most frequent trail users.

## Methods

### Eligibility criteria

Articles reporting injuries that occurred while people were mountain biking or hiking on trails were included, whether sustained by recreational or competitive participants. Hiking was taken to include all forms of pedestrian trail use including bush walking, orienteering trail running and hiking. Randomised control trials, cohort studies, case-control studies, case series and epidemiological studies were included; single case reports were excluded. Articles reporting injuries that occurred while cycling not on trails, such as roads or shared paths, were excluded, as were articles describing trail users who were utilising non self-powered transport such as motorbikes, horses or four-wheel drive vehicles. Studies without injuries reported by medical personnel, such as patient self-reported injuries, were excluded from the selection. Articles were required to have an English language abstract for initial screening and where deemed suitable for inclusions the manuscript was required to be translated into English for further assessment. One article was translated from French by a bilingual translator independent of the project prior to further assessment and data extraction.

### Outcome measure

The primary outcomes were types of injuries sustained by mountain bikers and hikers and their anatomical locations. A secondary outcome was the incidence of injury amongst mountain bikers and hikers.

### Data sources and search strategy

The search strategy was developed with advice from a medical faculty librarian and the study was prospectively registered with the University of York PROSPERO database on the 12/4/2021 (CRD42021229623). The following databases were searched (01/03/2022) for appropriate studies: CINAHL, Cochrane, ProQuest, Pubmed and Scopus. The search was undertaken independently by a single researcher (PJB) searching for peer-reviewed literature only. No published-date limitation was used, and there was no limitation on the age of the participants. Once all articles were identified, they were imported into Rayyan systematic review software, and duplicates were removed [30].

## Data screening and extraction

The remaining articles were then assessed for eligibility by two researchers, PJB and either PLB or TB to determine inclusion for a full-text review. Authors were blinded to each other's determination to avoid bias. A record of exclusions made at each step was recorded. Disagreement at this stage was settled by a third researcher (HT). Reference lists of the final articles were then also reviewed to identify any additional articles. Data were extracted by a single reviewer (PJB) and included study location, sex, age, types of injuries reported, anatomical location of injuries, injury incidence rate (where available) and mortality (where reported).

## Assessing study quality and bias risk

Overall study quality was assessed by two reviewers (PJB and HT) utilising the appropriate JBI critical appraisal checklists for each study design [31]. The JBI tools contain questions covering aspects of study design, risk of bias and reporting of results. No numerical rating was assigned to each study.

## Data synthesis

A descriptive narrative analysis was used to present the types of injuries sustained by mountain bikers and hikers. Anatomical locations of injuries reported by each study were extracted and are presented as descriptive tables (Tables 1 and 3). Descriptive tables showing the number and prevalence of injuries reported by each study and by each body location are also presented (Tables 2 and 4). To overcome the differences in reporting categories of different studies, anatomical locations for these injuries were grouped into four main areas; head, body, upper limbs and lower limbs. A meta-analysis was planned to compare anatomical locations of injuries both within activity types and between activity types if enough participants were found and the information was collected and reported in a homogenous manner.

# Results

## Search results

The search returned 10,095 studies for the title and abstract review, with a further three studies identified through snowball sampling of reference lists. Of the 10,098 articles, 4,436 duplicates were removed. After the title and abstract review, 134 articles remained. These were reviewed and 24 were selected after meeting the inclusion criteria (Fig 1).

All twenty-four included studies were assessed utilising the PRISMA checklist for case series and the results of the assessment are presented in S1 Checklist [31]. Only one study passed all assessment questions by both reviewers [32] with the remainder having at least one element missing. Of the 24 included studies, 17 involved mountain biking injuries [20,32–47] and seven involved hiking injuries [27,48–53]. The results describing mountain bike injuries are now considered separately to those relating to hiking injuries.

## Mountain biking

Of the 17 identified mountain biking studies, one study each was undertaken in Australia [41], France [37] England [42], Switzerland [45] and Scotland [46], four studies from Canada [32,36,40,44] and seven from the United States of America (USA) [20,33–35,38,39,47], with one additional study covering races across nine separate countries [43]. From these 17 mountain biking studies, eight investigated competitive racing [33–35,41,43,45–47] with the remaining nine not specifying the type of mountain biking being undertaken [20,32,35–40,42,44]. Seven competitive racing studies were based on data collected at races [33–35,41,43,45,47]

**Table 1. Descriptive statistics of included mountain biking studies.**

| First Author | Duration of study | Country | Injured patients (n) | Mean age (yrs) | Sex (male %) |
|---|---|---|---|---|---|
| Ashwell | 2009 | Canada | 898 | 26 | 86 |
| Cajani | 2015–2020 | Switzerland | 4920 | Unreported | 85 |
| Carmont | 2003 | Scotland | 52 | Unreported | Unreported |
| Chow | 1994–2008 | USA | 97 | 28 | 74 |
| Jeys | 12 months (date not provided) | England | 84 | 23 years | 83 |
| Kim | 1992–2002 | Canada | 399 | Unreported | Unreported |
| Kotlyar | 2012–2015 | USA | 304 | Any | 70 |
| Kronisch | 1995 | USA | 31 | Unreported | 79 |
| Kronisch | 1994 | USA | 16 | Unreported | 69 |
| Kronisch | 1994–2001 | USA | 156 | 28 (male) 31 (female) | 76 |
| Nelson | 1994–2007 | USA | 216707 | Unreported | 81 |
| Palmer | 2017–2018 | Multiple[*] | 179 | Unreported | 87 |
| Rivara | 1992–1994 | USA | 3390 | Unreported | 87 |
| Roberts | 1995–2009 | Canada | 49 | Unreported | 88 |
| Romanow | 2008–2010 | Canada | 409 | Unreported | 84 (cases) 89 (controls) |
| Saragaglia | 2014–2016 | France | 138 | 28 | 92 |
| Taylor | 2000–2007 | Australia | 596 | Unreported | 88 |

with the remaining ten studies being emergency department (ED) data [20,32,36–40,42,44,46,54]. Three cycling studies excluded children, one excluded children under 8 years [20], one excluded children under 16 years [32] and one excluded children under 17 years [43], with the remaining eleven including all ages however some were possibly limited by the competition minimum age. The patient characteristics for each mountain biking study are presented in Table 1.

Of the eight studies which examined competitive cycling, six recorded the exposed population as the number of competitors in the event [35,41,43,45–47]. Of these six studies, 1%-12% of female competitors were injured and 0% - 9% of male competitors were injured [35,41,43,45–47]. Of the remaining nine studies which did not report an exposed population, one was competitive mountain bike riding [33] and eight were non-competitive [20,32,35–40,42,44]. They reported between 8%-30% of the injured riders were female and between 70%-92% were male [20,32,33,35–40,42,44]. Two of the eight studies reporting competitive mountain biking injuries reported mean age distribution amongst injured riders; this was 28.3 and 28.4/30.2 (male/female) years [33,47]. Of the nine non-specified riding studies, mean age was reported as 15 [36], 19 [36], 23 [40], 28 [37] and 30 [20] and two report median age of injured riders as being

**Table 2. Injury locations reported in mountain biking studies.**

| First Author | Year | Country | Injured patients (n) | Age | Head | Face | Neck | Thorax | Abdomen | Upper extremities | Lower extremities |
|---|---|---|---|---|---|---|---|---|---|---|---|
| Ashwell | 2012 | Canada | 898 | Any | | X | X | X[3] | X[3] | X | X |
| Cajani | 2022 | Switzerland | 4920 | Any | X | | | X[3] | X[3] | X | X |
| Carmont | 2005 | Scotland | 52 | Any | X | | X | X | | X | X |
| Chow | 2002 | USA | 97 | Any | X[1] | X | X[1] | X[3] | X[3] | X | X |
| Jeys | 2001 | England | 84 | Any | X | | X | X | | X | X |
| Kim | 2006 | Canada | 399 | Any | X | X | X | X | X | X | X |
| Kotlyar | 2016 | USA | 304 | Any | X | | | X | X | X | X |
| Kronisch | 1996 | USA | 31 | Any | X[2] | X[2] | X[2] | X[3] | X[3] | X | X |
| Kronisch | 1996 | USA | 16 | Any | X[1] | X | X[1] | X[3] | X[3] | X | X |
| Kronisch | 2002 | USA | 156 | Any | | X | | X | X | X | X |
| Nelson | 2011 | USA | 216707 | ≥ 8 | X | X | | X[3] | X[3] | X | X |
| Palmer | 2020 | Multiple* | 179 | ≥ 17 | X | X | X | X[3] | X[3] | X[4] | X[4] |
| Rivara | 1997 | USA | 3390 | Any | X | X | X | X[3] | X[3] | X | X |
| Roberts | 2012 | Canada | 49 | ≥ 16 | X | | X | X | X | X | X |
| Romanow | 2014 | Canada | 409 | Any | X[2] | X[2] | X[2] | X[3] | X[3] | X | X |
| Saragaglia | 2020 | France | 138 | Any | X | | | X | X | X | X |
| Taylor | 2013 | Australia | 596 | Any | X[1] | X | X[1] | X | | X[4] | X[4] |

[1]Injuries to head and neck were reported combined.

[2]Injuries to head neck and face were combined.

[3]Injuries to thorax and abdominal region were not separated.

[4]Upper and lower extremities were not separated.

*Ireland, Madeira, New Zealand, Australia, Canada, Chile, Italy, France, Slovenia and Austria.

28 [32] and 33 [39] years old, (with the remaining two not reporting standard deviation but the range of ages as being 20–39 and 21–30 [38,40]). All studies reporting the average age of injured riders or most injured age group reported these to be under 40 years old.

## Injury types

All eight studies reporting on competitive mountain bike racing had similar inclusion criteria [33–35,41,43,45–47]. All eight studies included any patients who presented to the first-aid station or hospital. All eight of the studies reported all injuries found by medical personnel and were not limited to any injury severity; however, one study based on ED records did not report on the percentage of minor injuries [46]. Where all injuries were reported, abrasions/contusions and lacerations were the most common injury found, constituting 40% [45], 45% [43], 47% [41], 61% [33], 64% [47], 73% [34] and 74% [35] of all injuries. Fractures were a common occurrence in competitive racing and made up 1% [41], 3% [35], 11% [35], 17% [43], 18% [33] and 19% [47] of all injuries. Of the studies investigating mountain bikers in non-competitive events, six of these reported all injuries. These studies reported abrasions/contusions/lacerations to make up 8% [36], 28% [37], 39% [44], 42% [36], 45% [20], 57% [39] and 67% [38] of all injuries. Fractures constituted 15% [38], 26% [20], 30% [39], 34% [44], 37% [36], 42% [36] and 43% [37] of reported injuries. One study included only patients referred to a fracture clinic and found most fractures to be of the upper limbs [42]. A further study included only mountain bikers admitted to a regional trauma centre and they similarly found orthopaedic injuries to be the most common injury at 46% of all injuries [40]. This study also highlighted that 66% of all patients admitted required surgical intervention [40]. Roberts et al (2012) only included

**Table 3. Grouped injury locations in mountain biking studies.**

| First Author, year | Injuries to head, face and neck, n (%) | Injuries to torso, n (%) | Injuries to limbs, n (%)[a] | Injuries to upper limbs only, n (%) | Injuries to lower limbs only, n (%) |
|---|---|---|---|---|---|
| Ashwell, 2012 [44] | 10 (2) | 53 (12) | 384 (86) | 330 (74) | 54 (12) |
| Cajini, 2022 [45] | 24 (21) | 5(4) | 85 (75) | 46 (40) | 39 (34) |
| Carmont, 2005 [46] | 4 (7) | 6 (10) | 51 (84) | 27 (44) | 24 (39) |
| Chow, 2002 [33] | 39 (21) | 17 (9) | 134 (71) | 65 (34) | 69 (36) |
| Jeys, 2001 [42] | 14 (13) | 3 (3) | 92 (84) | 80 (73) | 12 (11) |
| Kim, 2005 [40] | 255 (30) | 328 (38) | 275 (32) | 148 (17) | 127 (15) |
| Kotlyar, 2016 [16] | 90 (54) | 12 (7) | 65 (39) | 56 (34) | 9 (5) |
| Kronisch, 2002 [47] | 1 (3) | 6 (18) | 26 (79) | 25 (76) | 1 (3) |
| Kronisch, 1996 [34] | 14 (23) | 3 (5) | 44 (72) | 21 (34) | 23 (38) |
| Kronisch, 1996 [34] | 10 (20) | 8 (16) | 31 (63) | 19 (39) | 12 (24) |
| Nelson, 2011 [20] | 34329 (17) | 31175 (15) | 143150 (69) | 100760 (48) | 42390 (20) |
| Palmer, 2020 [43] | 27 (15) | 22 (12) | 132 (73) | N/A | N/A |
| Rivara, 1997 [38] | 34 (19) | 17 (10) | 126 (71) | 72 (41) | 54 (31) |
| Roberts, 2011 [32] | 41 (43) | 47 (49) | 8 (8) | 8 (8) | 0 (0) |
| Romanow, 2014 [36] | 98 (22) | 38 (9) | 308 (69) | 243 (55) | 65 (15) |
| Saragaglia, 2020 [37] | 10 (6) | 13 (7) | 155 (87) | 117 (66) | 37 (21) |
| Taylor,[b] 2013 [41] | 60 (14) | 19 (4) | 353 (82) | N/A | N/A |

[a] Figures reported for limbs is a total of upper and lower limbs.

[b] This study reported a proportion of injuries as having the location 'multiple' and as such these were excluded from the analysis due to no specific location.

patients determined to be severely injured, defined as having an Injury Severity Score >12 [32]. This study reported the most common injury found in severely injured mountain bikers recorded in the trauma registry to be spinal fractures [32]. They also recorded 57% of mountain bikers to have head injuries and 40% of mountain bikers to have extremity fractures [32]. One study had a much larger data set than the others, recording 216,707 injured mountain bikers in the USA over 14 years [46]. In addition to the proportions of fractures and minor injuries reported, they also found over 10,000 cases of traumatic brain injury, accounting for nearly 5% of all recorded injuries [20].

**Table 4. Descriptive statistics of included hiking studies.**

| First Author | Duration of study | Country | Injured patients (n) | Mean age (yrs) | Sex (male %) |
|---|---|---|---|---|---|
| Chu | 2020 | Hong Kong | 48 | 53 | 29 |
| Faulhaber | 2006–2014 | Austria | 5658 | 58 (Fatal) 52 (Non-fatal) | 72 (Fatal) 55 (Non-fatal) |
| Gasser | 2009–2018 | Switzerland | 11220 | 59 | Not reported |
| McClean | 1989–1990 | Scotland | 517 | Not reported | Not reported |
| Scheer | 2010 | Spain | 39 | 44 | 70 |
| Vernillo | 2015 | Italy | 132[b] | 46 | 75 |
| Wong | 2003–2005 | Hong Kong | 275 | 39 | 58 |

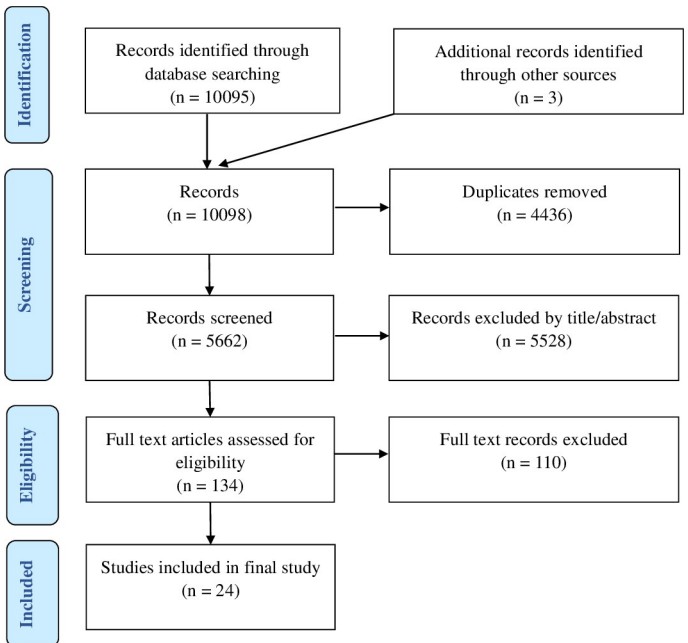

**Fig 1. Modified Preferred reporting items for systematic reviews and meta-analysis (PRISMA) flow diagram of included and excluded studies.**

## Mortality

No deaths were recorded in the eight studies investigating competitive mountain bike events [33–35,41,43,45–47]. Three studies investigating hospital records recorded one fatality each [32,38,40]. Head injuries were recorded as the cause of death for two fatalities [38,40] and the third fatality did not record a cause of death [32]. One fatality report of a head injury noted the patient was not wearing a helmet at the time of the incident [38].

## Injury incidence

Of the studies reporting on competitive mountain bike racing injuries, three studies reported incidence rate per hours of riding [35,41,43] and one reported incidence per number of starts [34]. For the studies reporting incidence per number of hours ridden, the reported incidences were 8 per 1,000 bike hours [41], 38 per 1,000 bike hours [43], 4 per 1,000 hours of competing in the cross-country event [35] and 43 per 1,000 hours of competing in the downhill event [35]. One study reported the overall incidence per number of starts to be 4 incidents per 1,000 starts, with the cross country event having an incidence rate of 3 per 1,000 starts and the down-hill event having an incidence rate of 5 per 1,000 starts [34].

## Injury location

There was considerable variation in how each study reported the location of injury. Of the 17 mountain biking studies, a summary of injury locations is presented in Table 2. For the purpose of this systematic review, the injuries reported were aggregated into three primary locations: injuries to the head, neck and face; injuries to the torso; and injuries to upper and lower limbs. These locations and number of injuries reported are presented in Table 3. As shown, injuries to the limbs are reported as the most frequent injury location in 14 of 17 studies and injuries to the torso being frequently reported as the least common injury location in 11 of 17

studies. Of the studies reporting on both upper and lower limb injuries, upper limb injuries dominated with 13 of the 15 studies reporting upper limb injuries to be more frequent. Injuries to limbs made up between 8%-86% of all injuries reported. Injuries to the torso were reported between 4% - 49% and injuries to the head, neck and face ranged between 6% - 54%. Injuries to upper limbs made up 8–74% of all injuries reported.

## Hiking

Of the seven studies reporting hiking data, three were either racing or competitive events [49,51,52], with three reporting on recreational hiking injuries from prehospital cases [27,48,50] and one based on hospital data [53]. Of the seven hiking studies, two were undertaken in Hong Kong [48,53] and one from each of Scotland [51], Italy [49], Spain [52], Austria [27] and Switzerland [50]. Descriptive statistics of all included hiking injury studies are presented in Table 4.

Of three studies examining competitive hiking, two were for trail running [49,52] and one for orienteering [51]. One trail running study reported 234 entrants and 77 (33%) of those agreed to take part in the injury study [49]. Of these 77 participants, 64 were male (83%) and 13 female (17%), with a mean age of 44 ±10 years of age [49]. The second study reporting trail running injuries reports on 69 participants of which 48 (70%) were male and 21 (30%) female [52]. The mean age of male runners was reported as 46 (27–63) years and for female runners 40 (26–50) years, though no SD was reported for either male or female runners [52]. The third competitive study examined orienteering across two separate events and reported demographic information for each [51]. Ages were not reported, however participants were categorised as junior or senior and male or female. The Loch Lomond event reported participants to be 9% junior female, 15% junior male, 26% senior female and 50% senior male [51]. At the Jan Kjellström event, the participants are reported as 8% junior female, 13% junior male, 25% senior female and 54% senior male [51].

One study examining aeromedical retrievals of hikers in Hong Kong reported that 159 (58%) evacuees were males and 116 (42%) females. The mean age was 39 with a range of 1–83, however no SD was provided. [48]. A further study on aeromedical evacuees of Austrian hikers found 53% to be female and 47% male [27]. The mean age of Austrian hiking evacuees was 53 (SD 17) [27]. Similarly, hikers admitted to a Hong Kong orthopaedic ward had a median age of around 51 years [53]. One study investigated all hikers calling for assistance in Switzerland between 2009–2018 and reported 11,200 calls for assistance from hikers [50]. Age was categorised into 10-year bins with 70–79-year-old hikers calling for assistance most frequently, accounting for 1,955 (20%) calls [50]. Hikers aged 60–69 years were the next most frequent with 1,858 (19%) of calls, overall hikers aged over 50 years were responsible for 70% of all call-outs [50].

Due to the high heterogeneity in the activities reported in the five studies, inter-study comparisons have not been made. Whilst all competitive studies reported on the sex of all participants, no study reported injuries by sex [49,51,52]. Additionally, whilst two of the studies about competitive trial use recorded the mean age of competitors, neither reported injuries by individual age or age group [49,51,52].

## Injury types

Blisters were reported as a common medical complaint in all three competitive studies [49,51,52], including during the Spanish Al Andalus trail run at 33% of all medical encounters [52], plus the most common skin complaint at the Italian Vigolana Trail® run at 54% of skin disorders and 10% of all injuries [49]. Blisters were not as common during the two orienteering events, constituting 8% and 15% of injuries [51]. Within all competitive events, the most

common injury during orienteering was reported to be ankle sprains, constituting 15% and 19% of all injuries [51]. During trail running, ankle sprains were recorded as the most common injury at one event with 29% of injuries [49] whilst at another event making up only 5% of injuries [52]. This event reported patellofemoral pain syndrome to be the most common injury, accounting for 9% of all injuries [51]. In non-competitive recreational hiking in Hong Kong, injuries were the most common reason for aeromedical evacuation accounting for 31% of all evacuations [48]. The most common injury (11%) was classified as soft tissue, with fractures being the second most common at 9% of cases [48]. Another report from Hong Kong included only patients admitted to the orthopaedic ward and, accordingly, fractures of the tibia/fibula were the most frequently reported injury [53]. In Switzerland, 4,900 cases contributing 44% of evacuations were caused by hikers falling [48]. No further details into the nature of the injuries sustained by falls is provided, however 15% of rescues were for head and skull trauma [50].

## Mortality

Of the two studies reporting on evacuation of injured hikers, both included reports of deaths [48,50]. A total of five deaths were recorded in hikers evacuated from Hong Kong over a three-year period, with three presumed cardiac in origin, one traumatic fall and one cause not recorded [48]. In Switzerland it was reported that an annual rate of 2 deaths per 2,000,000 hikers occurred over a 10 year period [50]. One study from Austria reported 331 fatalities over a nine year period, with a consistent number of deaths per year however, when considering death as a proportion of outcomes, mortality declined over the study period from 7.2% to 4.4% [27].

## Injury incidence

Orienteering was described as having 2% of all starts requiring first aid for injuries [51]. Trail running on the Vigolana trail (Italy) had an injury incidence rate of 1,886/1,000 runners or 13/1,000 hours running [49]. When considering non-competitive events, only one hiking study presented an incidence rate and this was for mortality, (2 deaths per 100,000 hikers per year) [50].

## Injury location

Of the seven studies reporting on hiking injuries, the injury location classification used in each is shown in Table 5.

There was considerable variation in how each study reported the location of injuries. Injuries to the limbs were the most dominant location in all studies and accounted for between 65–100% of all reported injuries (Table 6). Injuries to the head, neck or face were reported in five studies and were the second most common locations for injuries to hikers [48–51]. Injuries to the torso were less common and only reported to have occurred in five of the seven studies (Table 4) [27,48,50,51,53]. All studies reported on both upper and lower limb injuries but only five reported both having occurred (Table 6). All five reported lower limb injuries comprising between 49%-100% of all reported injuries [27,48,50,51,53]. Upper limb injuries accounted for between 0% - 30% of all reported injuries in these three studies [27,48,50,51,53].

## Discussion

This review is the first to systematically report injury types and anatomical injury locations between the two most common trail users, mountain bikers and hikers. There are four main findings to come out of this review: (1) injury types amongst mountain bikers were dominated by abrasions, lacerations and contusions with fractures of the shoulder girdle also common;

**Table 5. Injury locations reported in hiking studies.**

| First Author | Year | Country | Injured patients (n) | Age | Head | Face | Neck | Thorax | Abdomen | Upper extremities | Lower extremities |
|---|---|---|---|---|---|---|---|---|---|---|---|
| Chu | 2021 | Hong Kong | 48 | Any | | | | $X^2$ | $X^2$ | X | X |
| Faulhaber | 2017 | Austria | 5658 | Any | X | | | $X^2$ | $X^2$ | X | X |
| Gasser | 2019 | Switzerland | 11220 | Any | X | | | X | $X^4$ | X | X |
| McClean | 1990 | Scotland | 517 | Any | $X^1$ | $X^1$ | $X^1$ | $X^2$ | $X^2$ | X | X |
| Scheer | 2011 | Spain | 39 | Any[a] | | | | | | | X |
| Vernillo | 2015 | Italy | 132[b] | Any[a] | | | X | | | | X |
| Wong | 2010 | Hong Kong | 275 | Any | $X^3$ | $X^3$ | $X^3$ | $X^2$ | $X^2$ | X | X |

[1] Injuries to head neck and face were combined.

[2] Injuries to thorax and abdominal region were not separated.

[3] Injuries to head, neck, face, thorax and abdomen were combined.

[4] Injuries to abdomen were included within polytraumas.

[a] Age likely limited by competition but not stated.

[b] Study reported only total number of injuries not the number of injured participants.

(2) injury types amongst hikers were dominated by blisters and ankle sprains; (3) most injured mountain bikers sustained upper limb injuries and (4) most injured hikers sustained lower limb injuries. Many studies were excluded from this analysis due to injuries being self-reported or reported by non-medical personnel, or because they utilised questionnaires to gather injury data. The authors felt that the inherent bias involved in self-reported injuries and the inaccuracy of self-diagnosis made them unreliable sources of data. We included 24 studies in total, 17 on mountain biking and seven reporting on hiking injuries. The study designs within this review were predominantly prospectively collected or retrospective case series, with the majority being based on chart review of medical records. There was high heterogeneity in both injury definition, inclusion criteria, reporting mechanism and in estimating the exposed population. For these reasons, a meta-analysis was not deemed appropriate both within and between the two trail user groups. A recent systematic review investigating cross country mountain biking injuries was also unable to complete a meta-analysis due to the wide definition of injury [55], and the same was true for this systematic review.

**Table 6. Injury locations in included hiking studies.**

| First Author | Injuries to head, face and neck, n (%) | Injuries to torso, n (%) | Injuries to all limbs, n (%) | Injuries to upper limbs, n (%) | Injuries to lower limbs, n (%) |
|---|---|---|---|---|---|
| Chu | 0 (0) | 4 (9) | 40 (91) | 10 (23) | 30 (68) |
| Faulhaber | 688 (14) | 481 (10) | 3658 (76) | 446 (9) | 3212 (67) |
| Gasser [a,b] | (15) | (8) | (81) | (14) | (67) |
| McClean [c] | 82 (19) | 7 (2) | 337 (79) | 126 (30) | 211 (49) |
| Scheer | 0 (0) | 0 (0) | 55 (100) | 0 (0) | 55 (100) |
| Vernillio | 4 (6) | 0 (0) | 59 (94) | 0 (0) | 59 (94) |
| Wong | 21 (25) | 5 (6) | 57 (69) | 8 (10) | 49 (59) |

[a] Total is below 100% due to some injuries not having a location specified.

[b] Data reported as percentage only.

[c] Total is above 100% due to polytrauma including the individually recorded injuries.

## Mountain biking

Men were reported to be injured more frequently than women whilst mountain biking however, as many studies did not know the population denominators, it was not possible to determine if this was because more men undertake mountain biking than women. For studies that included a known population such as races, four reported a higher proportion of women to be injured [20,35,41,47], one reported more men to be injured, and one reported no significant difference [43,45].

For mountain bikers, the bodily region most affected was the lower limb, then injuries to the head, neck, and face were the next most common. This is consistent with the mechanism of injury, with many studies on mountain biking injuries reporting riders being thrown from the bicycle and utilising their upper limbs to protect themselves [42,56–58]. As mountain biking is often practised downhill, there is potential for riders to be travelling at speed when they crash, with an associated transference of kinetic energy through the limbs contacting the ground or object. This is supported by studies that investigated the circumstances of crashes which led to an injury, as they found riders who were thrown over the handlebars were more likely to sustain serious injuries compared with riders who fell from the side of a bike [20,33–35].

Helmet use amongst mountain bikers is extremely common with one study in this review finding over 99% of injured mountain bikers were wearing helmets [36]. Our review found injuries to the head, neck and face accounted for between 6–54% of injuries. Considering this, awareness of concussion may not be adequate amongst riders with one study reported 29% of competitive riders continuing to race with a broken helmet after a crash and 67% of riders who sustained a concussion continuing to ride [59]. A concussion protocol has been previously proposed as a necessary inclusion into mountain bike racing [43], and a recent consensus statement by the International Cycling Union recognised this requirement but highlighted the difficulty in implementing it in mountain biking and additionally called for a mountain biking specific concussion protocol [60].

Despite the potential for high severity of injury, most reported injuries were abrasions, contusions and lacerations in competitive mountain biking studies. One competitive mountain biking study reported fractures to be the most common injury; however, it was limited to patients admitted to a hospital ED, with a consequential likelihood of more severe injuries being present [46]. As most studies required mountain bikers to present at a first-aid facility or request assistance, it is possible that some riders self-treated their wounds, potentially leading to an overestimate in the proportion of severe injuries.

Fractures during competitive events were reported to constitute 2–18% of all injuries [33,35,41,43] and between 15–42% of all injuries in non-competitive mountain biking [36–39]. Several studies recommended clothing to protect against lacerations and abrasions [37,41]. The use of protective equipment is now common amongst downhill mountain bikers with a recent study reporting 86% of injured downhill mountain bikers wearing a full-face helmet, 76% wore elbow pads and 72% a protective vest [37]. The next step in protective equipment could be the development of suitable lightweight, breathable and protective outer clothing to minimise the potential for lacerations and abrasions.

The types and frequency of injuries reported were dependent on the inclusion criteria for an injured cyclist, and this varied between studies. For studies that had strict inclusion criteria, such as admission to a trauma centre and a high injury severity score, reported minor injuries were uncommon, most frequently being head and spinal injuries [32,40]. Injuries to the shoulder girdle were often reported, and these included fractures to the clavicle and scapula, in addition to dislocations of the shoulder. Where studies reported all injuries, upper limb fractures to this region were often the most reported, or second most reported, injury type

[20,37,39,42,44,47]. Indeed, one study found injuries to the shoulder and clavicle to have the greatest burden of days lost from work [43].

## Hiking

In contrast to mountain biking, most injuries reported by hikers were of the lower limbs, which were reported by all studies in this review (Table 4). The proportion of injuries to limbs was between 69–100% of all injuries, and the proportion of injuries to lower limbs was 49–100% [27,48–53]. The location of these injuries is again consistent with the mechanism of injury amongst hikers. For studies involving competitive hiking where all injuries were reported, we found that blisters and ankle sprains were the most often reported injury. A recent systematic review of trail running injuries found blisters to be common amongst trail runners [61]. Previous research has shown blisters to be present in 75% of hikers completing a long distance trail [62], in addition to being the most common complaint amongst trail runners [62]. Risk factors for the development of blisters include moisture, temperature, the duration of the activity and the type of footwear [63,64], and two of the studies involved ultra-marathons in hot climates. Previous research has reported the prevalence of ankle sprains in hikers to be around 9% and has suggested risk factors include loose ground and walking downhill, due to loading of the joint [65]. The studies we reviewed included only injured participants and ankle sprains constituted 5%, 15%, 19% and 29% of all injuries reported in three competitive hiking studies. Competitive hiking events such as trail running are undertaken at a fast pace, often on unknown trails; for this reason, it is likely that participants will be exposed to uneven surfaces with less time to react than during recreational hiking. Ankle sprains can lead to chronic ankle instability and an increase in the chance of further ankle sprains or falls [66]. Previous work has suggested that including ankle injuries in studies on acute injuries during trail running is problematic due to the prevalence of chronic overuse injuries in the ankles of trail runners [67]. This highlights a fundamental difficulty in acute sports injury epidemiology whereby it is often difficult to separate chronic overuse injuries from acute single event injuries [67].

A recent systematic review of injured trail runners found knee injuries to be a common occurrence [61]. We found concurring results with knee injuries being common in orienteering [51], recreational hiking [50], and trail running [52]. A prospective study in military recruits during hikes found 25% of all reported injuries were of the knee joint [64]. The presence of foot blisters was shown to be an important factor in the development of overuse knee injuries [64], possibly due to altered gait and compensatory mechanisms. Causes of knee injuries are believed to be similar to those of ankle injuries with uneven ground and the long exposure time of events such as ultra-marathon trail running. Additionally, a prospective cohort study investigating injuries amongst hikers with backpacks reported predominantly lower limb injuries of the leg, foot, knee and ankle [68]. They found that being overweight with a BMI > 30 kg/m$^2$ and/or having a higher backpack weight to body weight ratio was significantly more likely to lead to an injury [68]. As many of the injuries reported in hiking studies could be chronic overuse and not acute, these results may best be considered an overall picture of injury prevalence in trail users rather than injuries that occur suddenly during the activity.

## Strengths and limitations

Like previous work investigating both mountain biking and hiking injuries, the lack of a consistent definition of injury prevents the ability to compare results both within and between trail user groups [55,69]. Additionally, many studies reporting injuries to trail users were not included in this review due to the use of questionnaire-based injury identification and the

inherent recall and selection bias that comes with this type of study design [70,71]. For this reason, when calculating incidence rates of injury we were forced to rely on competitive studies where the exposed population was known from the number of competitors. Amongst studies with known exposed populations, the incidence rate was widely heterogenous with denominators including race hours, practice hours, hospital catchment, and race starts.

Despite these limitations, a strength of this review is to capture medically diagnosed injuries to the two main trail user groups which, to our knowledge, has not previously been undertaken. The inclusion of both prehospital event data and hospital records highlighted a wide range of injuries and captured people who may not have attended hospital with minor injuries. Whilst a meta-analysis was not possible for the aforementioned reasons, the predominance of upper limb injuries amongst mountain bikers and lower limb injuries amongst hikers cannot be discounted.

## Conclusions

Mountain bikers typically sustain contusions, lacerations, and abrasions. When considering more serious injuries, mountain bikers typically sustain injuries to the shoulder girdle including clavicle fractures and shoulder dislocations. The most commonly injured location were the upper limbs. Hikers' injuries were predominated by blisters of the foot, in addition to sprained ankles. Whilst ankle sprains may not be acutely life altering injuries, they have the potential to carry a high burden of disease due to the increased likelihood for repeated strains. In addition to ankle injuries, knee injuries were also common, especially amongst the more competitive hiking sports such as trail running and orienteering. Hikers predominantly sustain injuries to the lower limbs. A standardised method of reporting injury should be followed by future authors to enable cross caparisons between cohorts in addition to the reporting of all injuries to give a true indication of the prevalence of severe injury within these sports.

## Supporting information

**S1 Checklist. PRISMA checklist.**
(DOCX)

**S1 Table. Search strategy.**
(DOCX)

**S1 Appendix.**
(DOCX)

## Author Contributions

**Conceptualization:** Paul John Braybrook.

**Formal analysis:** Paul John Braybrook, Tanya Birnie, Deon Brink, Peter Buzzacott.

**Investigation:** Paul John Braybrook.

**Methodology:** Paul John Braybrook, Judith Finn.

**Project administration:** Judith Finn.

**Supervision:** Hideo Tohira, Deon Brink, Judith Finn, Peter Buzzacott.

**Validation:** Peter Buzzacott.

**Writing – original draft:** Paul John Braybrook.

**Writing – review & editing:** Paul John Braybrook.

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
