## [Decision Letter · Decision Letter 0]

14 Mar 2023

PONE-D-22-24910Types and anatomical locations of injuries among mountain bikers and hikers: a systematic reviewPLOS ONE

Dear Dr. braybrook,

Thank you for submitting your manuscript to PLOS ONE. After careful consideration, we feel that it has merit but does not fully meet PLOS ONE’s publication criteria as it currently stands. Therefore, we invite you to submit a revised version of the manuscript that addresses the points raised during the review process.

ACADEMIC EDITOR:  Dear corresponding Author, the paper is very interesting and it was well appreciated by the two reviewers. The final decisionof this first round of review is a minor revision. Please be careful to follow the comment of the Reviewers in order to improve you paper for the second round. I apologize for the delay and for the late but we had some troubles with a previous reviewer, now the process will be faster. 

We look forward to receiving your revised manuscript.

Kind regards,

Luca Russo, Ph.D.

Academic Editor

PLOS ONE

Journal Requirements:

Reviewers' comments:

Reviewer's Responses to Questions

**Comments to the Author**

1. Is the manuscript technically sound, and do the data support the conclusions?

Reviewer #1: Yes

Reviewer #2: Yes

2. Has the statistical analysis been performed appropriately and rigorously? 

Reviewer #1: Yes

Reviewer #2: Yes

3. Have the authors made all data underlying the findings in their manuscript fully available?

Reviewer #1: Yes

Reviewer #2: Yes

4. Is the manuscript presented in an intelligible fashion and written in standard English?

Reviewer #1: Yes

Reviewer #2: Yes

5. Review Comments to the Author

Reviewer #1: the manuscripts describe in detail the dussivision for the sport investigated, the types of injuries, depending on whether it is a competition or a non-competitive activity, based on gender and age. the resulting statistics are extrapolated from a large number of studies, Graphed in summary tables respecting all the criteria.

Reviewer #2: GENERAL and SPECIFIC COMMENTS:

Thank you for the opportunity to review this interesting issue and we value the effort that you put into this study. I think that this research is able to get the reader interested and work together on the topic. I think this article has some potential but there are some critical flaws as described below:

1. Both hikers and bikers are common trail users, but in a systematic review, it may be more worthy to study one group of people with a similar feature. In this aspect, choosing one type of participant or a variety of trail users, such as trail runners, is recommended in this study.

2. In the method section, I recommend using the risk of bias tools with scores, such as the modified Downs and Black assessment tool. The interrater reliability of raters is also recommended to report.

3. Please provide a summary table for the pooled studies and describe the characteristics, which is helpful for readers to get quick information.

4. The included studies contain the non-English study, please add more information about how the screening and data extraction for these articles was done and by whom.

5. Please do not use abbreviations for once-used sentences, such as EMS.

6. PLOS authors have the option to publish the peer review history of their article (what does this mean?). If published, this will include your full peer review and any attached files.

Reviewer #1: No

Reviewer #2: **Yes: **Haonan Wang

---

## [Author Response · Author response to Decision Letter 0]

12 Apr 2023

Dear Editor

As reviewer one did not have any comments that required addressing please find below the responses to reviewer number 2, Haonan Wang. We thank the reviewer for any astute comments and for taking the time to read the manuscript so fully. Overall, we agree with all the points raised as have made corrections as appropriate. These specific corrections are detailed below.

Thank you for the opportunity to review this interesting issue and we value the effort that you put into this study. I think that this research is able to get the reader interested and work together on the topic. I think this article has some potential but there are some critical flaws as described below:

1. Both hikers and bikers are common trail users, but in a systematic review, it may be more worthy to study one group of people with a similar feature. In this aspect, choosing one type of participant or a variety of trail users, such as trail runners, is recommended in this study.

Response - We thank you for your astute observation on this point. We also felt that only including mountain bikers and hikers would not represent all trail users. For this reason, we also included trail runners, hikers, orienteers, bushwalkers, and any other pedestrian trail users in the selection process. This was not made clear in the manuscript, and we have now addressed this to make it very clear that any articles covering pedestrian trail users and injuries were considered for inclusion. We did include papers on orienteering, trail running and hiking and feel that this, in conjunction with mountain bikers, covers all the major non-motorised trail users within Australia. Thank you for pointing this out. Now included under eligibility criteria is an explanation “Hiking was taken to include all forms of pedestrian trail use including bush walking, orienteering trail running and hiking.”

2. In the method section, I recommend using the risk of bias tools with scores, such as the modified Downs and Black assessment tool. The interrater reliability of raters is also recommended to report.

Response - We reviewed a number of available risk of bias tools and settled on utilising the JBI risk of bias tool over the downs and black. It was felt that with most studies being case series the JBI tool offered a better option due to having a specific tool for assessing these types of studies. However, reading back the methods section ‘Assessing study quality and bias risk’ it is clear I did not articulate that this tool assessed the risk of bias. This has been changed to better reflect that. We did not assign a numerical value as the JBI framework for assessing RoB does not assign a numerical value. After research into the various tools, we found evidence that assigning numerical values to RoB has inherent problems and can miss fundamental flaws in studies that other qualitative assessments do not (O’Connor, et al. 2015 - Failure of a numerical quality assessment scale to identify potential risk of bias in a systematic review: a comparison study ). I have also included the interrater reliability as per your suggestion.

3. Please provide a summary table for the pooled studies and describe the characteristics, which is helpful for readers to get quick information.

Response - This is a great idea and we appreciate your suggestion. Whilst each results section did include a summary table it did not include basic descriptive statistics such as the proportion of each sex injured and the mean age. This has now been included as two tables, one for mountain biking and one for hiking

4. The included studies contain the non-English study, please add more information about how the screening and data extraction for these articles was done and by whom.

Response - Thank you for this point you are correct that this information was missing and needs to be added to the manuscript. The article was screened initially as the abstract is available in English. It was then translated into French by a dual nationality bilingual French and English speaker who was independent of the project. For the data extraction portion, the translated tables and text were utilised. I have now included the following under the eligibility criteria. “Articles were required to have an English language abstract for initial screening and where deemed suitable for inclusion the manuscript was required to be translated into English for further assessment. One article was translated from French by a bilingual translator independent of the project prior to further assessment and data extraction.”

5. Please do not use abbreviations for once-used sentences, such as EMS.

Response - Thank you for pointing this out. We have reviewed the document and any instance where an abbreviation is only used once has been removed. This has reduced the number of abbreviations.

---

## [Decision Letter · Decision Letter 1]

27 Apr 2023

Types and anatomical locations of injuries among mountain bikers and hikers: a systematic review

PONE-D-22-24910R1

Dear Dr. braybrook,

We’re pleased to inform you that your manuscript has been judged scientifically suitable for publication and will be formally accepted for publication once it meets all outstanding technical requirements.

Kind regards,

Luca Russo, Ph.D.

Academic Editor

PLOS ONE

Additional Editor Comments (optional):

The Reviewers accepted the paper. Congratulation for your work.

Reviewers' comments:

Reviewer's Responses to Questions

**Comments to the Author**

1. If the authors have adequately addressed your comments raised in a previous round of review and you feel that this manuscript is now acceptable for publication, you may indicate that here to bypass the “Comments to the Author” section, enter your conflict of interest statement in the “Confidential to Editor” section, and submit your "Accept" recommendation.

Reviewer #1: All comments have been addressed

Reviewer #2: All comments have been addressed

2. Is the manuscript technically sound, and do the data support the conclusions?

Reviewer #1: Yes

Reviewer #2: Yes

3. Has the statistical analysis been performed appropriately and rigorously? 

Reviewer #1: Yes

Reviewer #2: Yes

4. Have the authors made all data underlying the findings in their manuscript fully available?

Reviewer #1: Yes

Reviewer #2: Yes

5. Is the manuscript presented in an intelligible fashion and written in standard English?

Reviewer #1: Yes

Reviewer #2: Yes

6. Review Comments to the Author

Reviewer #1: (No Response)

Reviewer #2: (No Response)

7. PLOS authors have the option to publish the peer review history of their article (what does this mean?). If published, this will include your full peer review and any attached files.

Reviewer #1: No

Reviewer #2: No

---

## [Editor Report · Acceptance letter]

8 Aug 2023

PONE-D-22-24910R1 

Types and anatomical locations of injuries among mountain bikers and hikers: a systematic review 

Dear Dr. Braybrook:

I'm pleased to inform you that your manuscript has been deemed suitable for publication in PLOS ONE. Congratulations! Your manuscript is now with our production department. 

Kind regards, 

on behalf of

Dr. Luca Russo 

Academic Editor

PLOS ONE